ecology/taxonomy and systematics/evolution

guppy, parasite, community, Trinidad and Tobago

**Author for correspondence:**
Jackie Lighten
e-mail: jackielighten@gmail.com

# Parasite diversity and ecology in a model species, the guppy (*Poecilia reticulata*) in Trinidad

Ryan S. Mohammed[1], Stanley D. King[2], Paul Bentzen[2], David Marcogliese[3,4], Cock van Oosterhout[5] and Jackie Lighten[6]

[1]Department of Life Sciences, The University of the West Indies, St Augustine, Trinidad and Tobago
[2]Biology Department, Dalhousie University, 1355 Oxford Street, Halifax, Nova Scotia, Canada B3H4R2
[3]Environment and Climate Change Canada, St Lawrence Centre, 105 McGill, Montreal, Quebec, Canada HY2 2E7
[4]St Andrews Biological Station, Department of Fisheries and Oceans Canada, 125 Marine Science Drive, St Andrews, New Brunswick, Canada E5B 0E4
[5]School of Environmental Sciences, University of East Anglia, Norwich NR4 7TJ, UK
[6]Biosciences, University of Exeter, Stocker Road, Exeter EX4 4PY, UK

CvO, 0000-0002-5653-738X; JL, 0000-0002-4228-1037

The guppy (*Poecilia reticulata*) is a model species in ecology and evolution. Many studies have examined effects of predators on guppy behaviour, reproduction, survival strategies, feeding and other life-history traits, but few have studied variation in their parasite diversity. We surveyed parasites of 18 Trinidadian populations of guppy, to provide insight on the geographical mosaic of parasite variability, which may act as a source of natural selection acting on guppies. We found 21 parasite species, including five new records for Trinidad. Spatial variation in parasite diversity was significantly higher than that of piscine predators, and significant variation in parasite richness among individuals and populations was correlated with: (i) host size, (ii) snail species richness, and (iii) the distance between populations. Differences in parasite species richness are likely to play an important, yet underestimated role in the biology of this model species of vertebrate ecology and evolution.

# 1. Introduction

The guppy (*Poecilia reticulata*) is a model organism of vertebrate ecology and evolution with a native range spanning northern

**Figure 1.** Trinidadian localities where guppies (*Poecilia reticulata*) were collected (electronic supplementary material, table S1). Pie charts illustrate parasite species observed in populations. Caroni Drainage is coloured turquoise, North Slope Rivers coloured green, and the unsampled Oropouche Drainage is orange. Las Cuevas (LC), Yarra 2 (Y2), Marianne River (M16, M3), Petite Marianne (PM), Madamas (Mad), Paria 7 (P7), San Souci (SS), Mission (Mis), El Cedro 2 (El2), Guanapo Quarry (GQ), Lopinot (Lop), Lower Arima (Ar.L), Lower Aripo (Ap.L), Santa Cruz (SC), Upper Arima (Ar.U), Upper Naranjo (Nar.U), Blue Basin (BB).

South America to Trinidad and Tobago, and a near global tropical distribution facilitated by human introductions. Much research has focused on natural guppy populations in Trinidad's Northern Range, comprising discrete and diverse riverine ecosystems in three major drainage basins: Caroni, North Slope and Oropouche (figure 1), each with a drainage-specific lineage of guppy [1]. Within and among drainages, guppies show significant variation in genetics, morphology, coloration, life-history, predator avoidance and reproduction [1–6].

The research of guppies in Trinidad has primarily focused on the interplay of predation and sexual selection, while studies on parasite-mediated selection are relatively underrepresented. As with predation, parasite-mediated selection pressures can drive parallel evolution in guppy coloration, behaviour and life history among populations [7–9]. Colour expression may be negatively correlated with parasite infection [10,11], i.e. parasitized male guppies devote less energy to mating and courtship and display less intense mating colours, thereby appearing less fit and attractive to females [10,12].

To date, few studies have examined the overall parasite communities of natural guppy populations, with full necropsies [4]. Given the extremely high levels of major histocompatibility complex (MHC) immune gene polymorphism in guppies (greater than 500 MHC class IIB alleles in Trinidad alone) [7,13–15], the many studies showing balancing selection on MHC, and correlations between genetic diversity and parasite richness [16–18], it is likely that parasite-mediated selection is important in wild guppy populations. To aid future studies aiming to explore parasite-mediated selection, we examine the effects of biotic (host size, predation regime, snail diversity), and abiotic factors (distance between populations) on parasite species richness infecting guppies.

## 2. Methods

Male guppies (*n* = 270) were collected from 18 populations in Trinidad in March 2016 (figure 1; electronic supplementary material, table S1), across the North Slope and Caroni Drainages. Only males were studied due to forming part of a larger study examining the relationship between parasites, host coloration and genetics. Populations were categorized as high or low predation (HP/LP), given the predatory fish assemblages known to affect guppy ecology and evolution [19–21]. This is a commonly used category in guppy studies, and piscine predators were confirmed upon sampling of established sites or assessed at new sites. The predatory fish fauna at each site was sampled by seine netting of 50 m stretch of river, and deployment of a cast net (4 m diameter) with equal sample effort of 10 throws. Adult guppies were corralled using a 1 m² seine net and scooped into a bucket of river water while still in the river, avoiding direct contact between fish and seine to prevent potential displacement of ectoparasites. Individual fish were then transferred to 375 ml disposable plastic cups containing approximately 100 ml of river water treated with stress coat (1 ml per 2 l of water). Cups were sealed with cellophane secured with elastic bands and stacked in buckets for transportation back

to the William Beebe Tropical Research Station, where guppies were held live until examination (within 24 h). Euthanized guppies were measured and examined using stereo- and compound microscopes for parasites, occupying the body, fins, eyes, buccal cavity, gills, heart, liver, gall-bladder, gonads, intestine, spleen and viscera/mesentery. Gyrodactylids were identified to the species level using microscopy. We assessed the potential of ectoparasites dislodgement by transportation to the laboratory and use of stress coat (which augments skin mucus production) by visually inspecting the water that guppies were held in. For 20 individuals across four populations the water samples were left to settle for circa 1 h (time taken for each dissection) after removal of each guppy. Circa 90% of the water was carefully removed from the container and the remainder was visually inspected under a light microscope for dislodged gyrodactylids. We found that the number of dislodged gyrodactylids was either zero or negligible, and so concluded that our method of sample collection did not unduly affect parasite quantification. Metacercariae of some digeneans could not be identified to species but were distinguished based on the assumption that taxa could be delineated by specialization through infection site, morphological features (e.g. eye spots, body size and shape) and if unique to a population. Lumping provisional species into fewer taxa did not alter the main ecological results described below (other than parasite richness). Additionally, aquatic snail species richness was evaluated at each site by random deployment of a 0.25 m squared quadrat eight times along the 50 m sampling stretch.

## 2.1. Parasite and snail species richness

We compared infracommunity (parasites of an individual) and the component community (parasites of host population) among two drainage basins, nine rivers and 18 populations (locality). A Kruskal–Wallis test assessed whether parasite species richness varied significantly between drainages and predation regime. A general linear model (GLM) was used with the number of parasite species infecting an individual as response variable, population nested within drainage as factors, and fish standard length as covariate. We also analysed the variation between guppies and the number of *G. bullatarudis*, the most common parasite in our study. We used a GLM with 'populations' as factor and 'standard length' of males as covariate, and the number of *G. bullatarudis* was $\ln(x + 0.1)$ transformed.

We furthermore analysed the presence/absence of parasite species at a population level, using 'population' as the degrees of freedom in a binomial mass function. Pearson's correlations were used to test whether the number of a given species of parasite is correlated to the number of parasites of another species, and also whether the number of digenean parasite species infecting individuals, which use snails as an intermediate host in their life cycle, was correlated to snail species richness within a population. A Mann–Whitney test was used to compare the snail species richness between drainages.

We examined the relationship between parasite richness and physical distance between population pairs, to test if parasites are transmitted easier across smaller distances. Physical distance (m) was measured as a straight line between sampling sites because many parasites that were observed complete their life cycle in birds, and so can be dispersed across land. We acknowledge that our approach does not fully account for the influence of topography, and unidirectional movements of dispersing guppies predominantly downstream; however, our approach is valid given that parasites infecting guppies can be dispersed via birds, other and fish species [22,23], which can also readily disperse over land and drainages.

We examined the relationship between Jaccard's community dissimilarity index (presence/absence), using *vegan* [24] in R [25], and the physical distance between two populations, and compared this relationship between the Caroni Drainage populations and between North Slope populations. We then used a GLM to test whether physical distance and drainage explain significant variation in the Jaccard's index between two populations. Here we used physical distance as the covariate, with drainage as fixed factor to examine similarity of regression lines slopes. Finally, we calculated the linear regressions between Jaccard's index and physical distance for both drainages.

Fish were collected under the ethical approval of The University of the West Indies, a research permit from Fisheries Division, Ministry of Agriculture, Land and Marine Resources, and a collection permit from Wild Life Section, Ministry of Agriculture, Land and Marine Resources granted to Dr Mohammed.

# 3. Results

In 270 mature male guppies (mean length $21 \pm 2.1$ mm, range 11–30 mm) examined from Trinidad's Caroni Drainage ($n = 135$), and North Slope ($n = 135$) (electronic supplementary material, table S1), 21

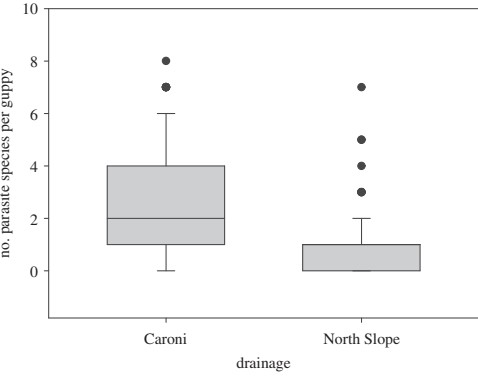

**Figure 2.** The number of different parasite species on an individual host is significantly higher on guppies in the Caroni Drainage than in the North Slope (Kruskal–Wallis test: $H = 75.62$, d.f. = 1, $p < 0.0001$).

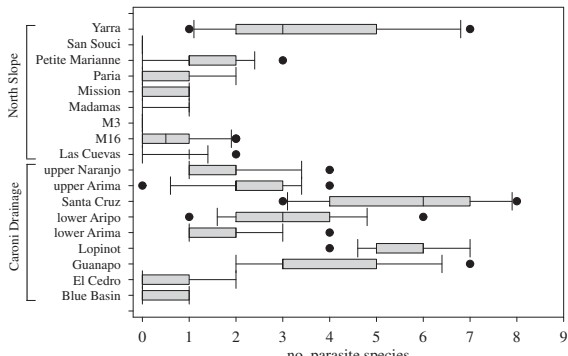

**Figure 3.** The number of different parasite species on an individual host differs significantly between populations (Kruskal–Wallis test: $H = 200.81$, d.f. = 17, $p < 0.0001$). North Slope populations are shown in the top part of the figure, the Caroni Drainage populations in the bottom part.

parasite species were found (electronic supplementary material, table S2), of which *Ascocotyle* sp., *Pygidiopsis* sp., *Posthodiplostomum* sp., *Saccocoelioides* sp. and *Myxobolus nuevoleonensis* represent new records for Trinidad. Parasite assemblages were dominated by digeneans (57% of all infections, 13 species, 12 of which were metacercariae), followed by monogeneans (42% of all infections, three species). The most abundant parasite was *Gyrodactylus bullatarudis*, followed by *Pygidiopsis* sp. Together, the three identified species of metacercariae (*Ascocotyle* sp., *Posthodiplostomum* sp., *Pygidiopsis* sp.), which mature in piscivorous birds, were found in 94/270 (34.8%) of the fish and 11/18 (61%) of the localities.

Parasite richness varied between drainages, and individuals in the Caroni Drainage were infected with significantly more parasite species (mean $2.38 \pm 2.03$) compared to those in the North Slope (mean $0.81 \pm 1.16$) (Kruskal–Wallis test: $H = 75.62$, d.f. = 1, $p < 0.0001$, figure 2). Caroni Drainage guppy populations showed on average a higher parasite richness than those from the North Slope (Kruskal–Wallis test: $H = 200.81$, d.f. = 17, $p < 0.0001$, figure 3). The number of parasite species on an individual did not differ between HP or LP populations (Kruskal–Wallis test: $H = 1.96$, d.f. = 1, $p = 0.161$). However, standard length explained a small amount of variation in parasite richness, with larger males infected with more species (GLM, $F_{1,240} = 13.41$, $p < 0.001$, table 1, electronic supplementary material, figure S1).

Seven parasite species were significantly more abundant in Caroni Drainage populations than in North Slope populations (i.e. *Posthodiplostomum* sp., *Saccocoelioides* sp., *Digenean* sp. 1, *Digenean* sp. 2, *Gyrodactylus bullatarudis*, *Gyrodactylus turnbulli* and *Trichodina* sp.; all binomial tests after Bonferroni correction: $p \leq 0.00504$, electronic supplementary material, table S3). None were significantly more abundant in the North Slope. A strong positive correlation existed between the abundances of parasite species across the Caroni Drainage (electronic supplementary material, table S4a), and within Caroni populations (electronic supplementary material, table S5), but not in the North Slope (electronic supplementary material, table S4b). These observations led us to test two further hypotheses: (i) the Caroni Drainage populations have a significantly higher snail species richness that can support a

**Table 1.** The GLM shows that population nested within drainage, drainage and standard length all explain significant variation in the number of different parasite species on an individual host.

| source | d.f. | MS | F | p |
|---|---|---|---|---|
| population (drainage) | 16 | 31.526 | 40.89 | <0.001 |
| drainage | 1 | 215.349 | 279.30 | <0.001 |
| length | 1 | 10.336 | 13.41 | <0.001 |
| error | 240 | 0.771 | | |
| total | 258 | | | |

**Table 2.** The GLM shows that both distance, and drainage explain significant variation in the Jaccard's dissimilarity index between two populations. However, the significant interaction term (physical distance × drainage) shows that this relationship is not the same across drainages.

| | d.f. | MS | F | p |
|---|---|---|---|---|
| physical distance | 1 | 0.4435 | 18.73 | <0.001 |
| drainage | 1 | 1.9620 | 82.87 | <0.001 |
| physical distance × drainage | 1 | 0.3440 | 14.53 | <0.001 |
| error | 68 | 0.0237 | | |
| total | 71 | | | |

higher parasite richness, and (ii) distance between guppy populations in the Caroni Drainage metapopulation facilitates parasite migration, and a higher per-population parasite richness compared to the North Slope.

Indeed, snail species richness was significantly higher in the Caroni Drainage populations than in the North Slope populations (Mann–Whitney test, $W = 61.5$, $p = 0.0380$). We furthermore found that guppies in populations with a higher snail species richness were infected with significantly more species of digeneans (Pearson correlation: $r = 0.563$, $p < 0.001$), and that this correlation was observed both in the North Slope (Pearson correlation: $r = 0.313$, $p < 0.001$), as well as in the Caroni Drainage (Pearson correlation: $r = 0.421$, $p < 0.001$, electronic supplementary material, figure S2).

Physical distance and drainage explain significant variation in Jaccard's community dissimilarity index between pairs of populations (table 2). The significant interaction term 'physical distance' × 'drainage' (GLM: $F_{1,68} = 14.53$, $p < 0.001$) shows that this relationship between parasite community and distance is not the same in both drainages. Figure 4 shows that Caroni Drainage populations that are close together share a relatively similar parasite fauna (regression: $F_{1,34} = 13.34$, $p = 0.001$, $R^2$ adj = 26.1%), but no such relationship exists for North Slope populations (regression: $F_{1,34} = 2.07$, $p = 0.159$, $R^2$ adj = 3.0%). Furthermore, figure 4 shows that the parasite communities associated with North Slope guppies are more distinct from each other than those of the Caroni Drainage. Importantly, the spatial variation in parasite diversity was significantly higher than that of piscine predator diversity (Kruskal–Wallis test: $H = 17.62$, d.f. = 7, $p = 0.013$, electronic supplementary material, table S6) as was that of snail species richness (Kruskal–Wallis test: $H = 17.99$, d.f. = 7, $p = 0.011$, electronic supplementary material, table S7), but dissimilarity based on snails was not significantly different from that based on parasites (Kruskal–Wallis test: $H = 51.06$, d.f. = 38, $p = 0.076$).

## 4. Discussion

Almost 80 species of parasites are reported from guppies, mostly in captive or experimental settings (electronic supplementary material, table S8). In wild guppies across the globe, approximately 30 parasite species are reported, and only 10 have been found in Trinidad. However, we detected 21 parasite species in just 270 guppies in 18 populations in Trinidad, including five species not previously recorded on any fish species from the island. Predation regime had no effect on parasite

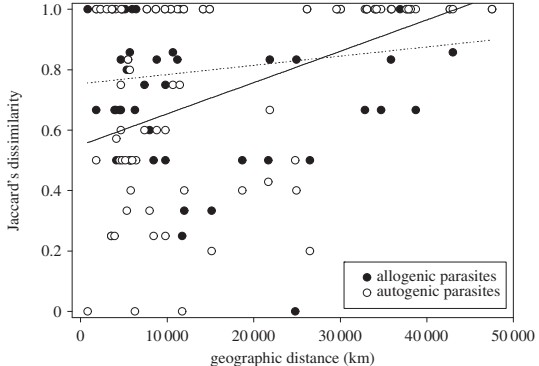

**Figure 4.** The relationship between pairwise population distance and dissimilarity in their parasite communities. Caroni Drainage population guppies show a significant positive association (regression: $p < 0.001$, $R^2$ adj $= 26\%$) between Jaccard's dissimilarity of parasites and the physical distance. No such relationship is detected for the North Slope guppies.

species richness, and populations were more differentiated by parasite species richness than piscine predator richness. We identified three factors which correlate with variation in parasite species richness between individuals, and populations; (i) host size, (ii) the snail species richness, and (iii) distance between populations.

## 4.1. Standard length

At the individual level, parasite species richness increased with standard body length, and parasite abundances were significantly correlated. Increased species richness with fish size has been observed in numerous freshwater fishes [26], presumably because larger (and older) hosts eat more, have a broader diet and have accumulated parasites over time [26,27]. Moreover, an increased surface area, has been shown to be positively correlated to the number of parasites on fish [8,27,28].

## 4.2. Population proximity

Strikingly, parasite richness was significantly higher in Caroni Drainage populations. Yet, North Slope populations showed more heterogeneity in parasite communities. North Slope guppy populations were sampled from separate rivers, or are separated by waterfalls that obstruct migration (e.g. a greater than 20 m waterfall separates the Petite Marianne and Marianne River). Conversely, the Caroni Drainage populations were sampled in a single interconnected metapopulation (excluding Blue Basin in the Diego Martin River). Geographical distance played a role in structuring guppy parasite communities, with populations nearby sharing more parasite species and similar infection levels. Within the Caroni, the strong relationship between parasites and distance may be explained by the interconnected nature of this metapopulation [29]. However, as we only measured the minimum distance between populations in a straight line, a more detailed analysis of connectivity is required to explain if parasites more readily disperse between populations through guppies or other intermediate hosts. The greater connectivity between populations of guppies and parasites in the Caroni Drainage may explain why seven out of the 21 parasites species detected were significantly more common in the Caroni. In contrast, the North Slope has increased isolation of populations in distinct rivers (figure 1), which may also affect parasite species richness. Indeed, population connectivity was found to be important in structuring parasite communities of other fish species [30–32].

## 4.3. Snails

The difference between drainages is further amplified by higher snail species richness in the Caroni Drainage. Seven species of snails (*Melanoides tuberculata*, *Gundlachia* sp., *Physella acuta*, *Pomacea glauca*, *Pseudosuccinea columella*, *Pyrgophorus parvulus* and *Biomphalaria* sp.) were found only in the Caroni. Only *Neritina virginea* was unique to the North Slope. Given that snails are an intermediate host for digeneans, and that snail species richness is positively correlated with parasite species richness (both within and between drainages), the higher species richness of snails probably elevates fish parasite

richness in the Caroni. Similarly, the diversity of benthic invertebrates acting as secondary parasite hosts was a primary contributor to parasite community composition in the common killifish (*Fundulus heteroclitus*) [33].

# 5. Conclusion

We identified three factors that correlate with variation in species richness of guppy parasites in the Caroni Drainage, and North Slope in Trinidad: (i) host size, (ii) snail species richness, and (iii) distance between populations. We have also made detailed observations specific to particular parasite species, and provided a summary of all known parasites infecting guppies (electronic supplementary material). Studies of natural selection on one of the best-known ecological models may benefit from examining selective forces exhibited by communities of parasites, as besides predator-mediated selection, parasite could also be important in guppy evolutionary ecology, as reflected by Haldane in that '… it is much easier for a mouse to get a set of genes which enable it to resist *Bacillus typhi murium* than a set which enable it to resist cats' [34].

Ethics. During the fieldwork, we collected fish under ethical approval granted from The University of the West Indies for R.S.M.'s PhD study, which allowed collection of *Poecilia* sp. (10 000 individuals maximum) over a 5-year period from 2012 to 2017 from waterways across Trinidad and Tobago. The Research Permit came from Fisheries Division, Ministry of Agriculture, Land and Marine Resources. The Collection Permit came from Wild Life Section, Ministry of Agriculture, Land and Marine Resources.

Data accessibility. All data are available in the electronic supplementary material.

Authors' contributions. R.S.M. contributed to fieldwork design, sample and metadata collection, and provided critical input towards the draft manuscript. S.D.K. collected all parasite data in the field and from the literature, and contributed to drafting the early manuscript. P.B. and D.M. provided resources/reagents/materials and critical input and discussion throughout the project. C.v.O. conceived the study, performed analyses and wrote the paper. J.L. conceived the study, performed analyses, led fieldwork and wrote the paper. All authors read and approved the final manuscript.

Competing interests. We declare we have no competing interests.

Acknowledgements. Funded by a British Ecological Society Large Research Grant (5875-6919) awarded to J.L. We are grateful to Tomáš Scholz, Thomas Fayton and Leo Aguirre Macedo for aiding in trematode identification. We thank Ronnie Fernandez for hosting fieldwork at the William Beebe Tropical Research Station.

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
