## [Reviewer comments · Royal Society Open Science]

Review History

RSOS-191112.R0 (Original submission)

Review form: Reviewer 1

Is the manuscript scientifically sound in its present form?

Yes

Are the interpretations and conclusions justified by the results?

Yes

Is the language acceptable?

Yes

Do you have any ethical concerns with this paper?

No

Have you any concerns about statistical analyses in this paper?

No

Recommendation?

Accept as is

Comments to the Author(s)

The authors present a field study on parasite load in guppies - a, as they point out, so far understudied factor in guppy selection. They provide very detailed data on individual infections and population infestation and show compelling evidence for what influences parasite load in different populations. It becomes clear that parasites play a major role in guppy ecology. The manuscript is well written and provides a large amount of interesting data (mainly in the supplements). The statistical analyses seems appropriate and the conclusions drawn from the results are convincing. I can therefore recommend publication. For Figure 1 I would appreciate a legend explaining the color code of parasites in the pie charts.

Review form: Reviewer 2**Is the manuscript scientifically sound in its present form?**

No

Are the interpretations and conclusions justified by the results?

No

Is the language acceptable?

Yes

Do you have any ethical concerns with this paper?

No

Have you any concerns about statistical analyses in this paper?

No

Recommendation?

Major revision is needed (please make suggestions in comments)

Comments to the Author(s)

Review King et al.

King et al. report parasite richness data for male guppies sampled from 18 populations in their native range in Trinidad. They found 21 species of parasites (most of them identified to the genus level) and show significant differences in parasite richness at the individual and population levels. Their data shows that individual size, snail richness and spatial distance among populations correlates with parasite richness. I think this paper will be of interest to the large community of researchers working on guppies. I found tables S9 and S3 particularly valuable, although I was a bit disappointed that many of the parasites were not identified to the species levels (I do recognize that doing so can be extremely challenging).

Major comments**-Selection**

Throughout the manuscript, particularly in the conclusion, the authors make claims about the links between parasite richness and selection in the guppy system. Beyond the reasonable expectation that these different parasites (e.g. the most abundant or virulent ones) could influence selection of host resistance, other claims are highly speculative and should be clearly flagged as

such. The data does not support these claims since no life-history, behavioural, physiological or other traits of the hosts were measured. Furthermore, the fact that parasite richness is higher than predator richness does not suggest that parasites are a stronger source of selection; I suggest removing such statements.

-Distance and connectivity

Connectivity and distance between populations are used interchangeably in the text (e.g. paragraph starting at L215). These two are not the same, particularly in streams. For example, given that the Northern Slope streams are all independent drainages they are not interconnected (no connectivity) and what the spatial distance between populations implies is not straightforward. The Caroni basin has instead all streams interconnected (although in a directional manner, which does not seem to be accounted for in the analyses) and distance between populations can have clear cut implications for parasite distributions. I suggest the authors clarify what they mean by distance and interconnectedness, and provide clear arguments for using these measures in streams that don't have a common freshwater tributary (Northern Slope). Furthermore, the authors should clarify how they measured distance, am I right to assume that they simply measured distance over a map using a straight line between populations? This would be problematic since it can be a poor reflection of direction of flow and the actual distance fish or parasites would have to travel among populations, and assumes topography plays no role in the movement of parasites and hosts.

-Sampling effort, snail and predator richness

The authors present correlations between parasite richness and predator and snail richness which form an important component of their rationale for the paper and their discussion (e.g. L65-68, L233-243). Nonetheless how the information on predator and snail richness was gathered is omitted (predators) or vaguely stated (snails). Was the sampling effort at all sites comparable? How exhaustive were the methods to detect these other species? Understanding this would help ease concerns when evaluating results such as those in Figure S2, where the regression seems to be strongly influenced by two sites with no snails, or Figure S3, comparing population-pair community dissimilarities in predators or parasites.

-Digenean instead of parasite correlations with snail richness

The authors formulate a hypothesis linking snail richness to overall parasite richness. It would be more adequate to present correlations between snail-transmitted parasites and snail richness only (the rationale here is clear, unlike mechanisms linking *Trichodina*, *Camallanus* or *Gyrodactylus* to snail richness).

-Parasite species identification

I suggest the authors explain further how different species of *Digenea* sp. 1-9 were identified (and possibly add images as supplementary materials). Knowing that there are 9 species present is of limited use, whereas researchers working in Trinidad could benefit from being able to distinguish among those species. Similarly, please clarify whether all *Gyrodactylus* were identified by location on the host or via microscopy? These methods differ substantially in reliability.

-Correlations

Given the nature of the study (field data collection) the results can only support correlations between certain variables and parasite richness, therefore the host size, snail richness and distance among populations cannot "explain" parasite richness. This should be revised throughout the manuscript.

Minor comments

-I suggest the authors use the term "richness" instead of "diversity" / "biodiversity", particularly in the title and abstract, as this is the only measure of diversity in this study.

- L25 -Be specific about “effects of predators on guppy biology”
- L26 -Clarify how this is going to aid studies of natural selection.
- L37 -Clarify model organism for what.
- L84-85 -More information is needed - describe the protocol, sampling effort and what did ‘supplementing’ consisted on.
- L90 -‘Wallis’ not “Wallace”.
- L94 -Clarify what kind of variation.
- L112 -Physical distance can interpreted in many ways, please be very clear on how distance was measured.
- L210 -Clarify what wider range implies: diets, habitats?
- L250-251 -This claim is not supported by the data, and is not a claim that can be reasonably made based on differences in richness.
- L251-253 -This is highly speculative and overstates the findings.
- L258 -“RH” is not one of the co-authors or is it a typo? Please clarify (1) what was their contribution, and (2) how the fish used in this study were chosen out of the 10,000 that were potentially sampled. If RH is not one of the co-authors please properly acknowledge their contribution.

Supplementary Materials

-Information under the subheading “Sampling” explaining the rationale for using males and guppy collection procedure is better suited for the main text rather than as a supplementary material. While reading the manuscript I wondered why the authors had only used male guppies and whether the sampling protocol increased parasite transmission.

Decision letter (RSOS-191112.R0)

29-Jul-2019

Dear Dr Lighten,

The editors assigned to your paper ("Parasite diversity and ecology in a model species, the guppy (*Poecilia reticulata*) in Trinidad") have now received comments from reviewers. We would like you to revise your paper in accordance with the referee and Associate Editor suggestions which can be found below (not including confidential reports to the Editor). Please note this decision does not guarantee eventual acceptance.

Please submit a copy of your revised paper before 21-Aug-2019. Please note that the revision deadline will expire at 00.00am on this date. If we do not hear from you within this time then it will be assumed that the paper has been withdrawn. In exceptional circumstances, extensions may be possible if agreed with the Editorial Office in advance. We do not allow multiple rounds of revision so we urge you to make every effort to fully address all of the comments at this stage. If deemed necessary by the Editors, your manuscript will be sent back to one or more of the original reviewers for assessment. If the original reviewers are not available, we may invite new reviewers.

When submitting your revised manuscript, you must respond to the comments made by the

referees and upload a file "Response to Referees" in "Section 6 - File Upload". Please use this to document how you have responded to the comments, and the adjustments you have made. In order to expedite the processing of the revised manuscript, please be as specific as possible in your response.

- Data accessibility

If you wish to submit your supporting data or code to Dryad (<http://datadryad.org/>), or modify your current submission to dryad, please use the following link:
<http://datadryad.org/submit?journalID=RSOS&manu=RSOS-191112>

- Competing interests

- Authors' contributions

- Acknowledgements

- Funding statement

on behalf of Dr Michael Tobler (Associate Editor) and Kevin Padian (Subject Editor)
openscience@royalsociety.org

Associate Editor's comments (Dr Michael Tobler):

We have received the feedback from two reviewers. Overall, both agree that the paper is meritorious. However, one reviewer also provided some excellent feedback on how to potentially improve the current version of the manuscript. Pending satisfactory revisions, this manuscript can be suitable for publication in RSOS. I would also like to point out that a condition of publication is that authors make their supporting data, code and materials available.

Comments to Author:

Reviewers' Comments to Author:

Reviewer: 1

Comments to the Author(s)

The authors present a field study on parasite load in guppies - a, as they point out, so far understudied factor in guppy selection. They provide very detailed data on individual infections and population infestation and show compelling evidence for what influences parasite load in different populations. It becomes clear that parasites play a major role in guppy ecology. The manuscript is well written and provides a large amount of interesting data (mainly in the supplements). The statistical analyses seem appropriate and the conclusions drawn from the results are convincing. I can therefore recommend publication. For Figure 1 I would appreciate a legend explaining the color code of parasites in the pie charts.

Reviewer: 2

Comments to the Author(s)

Review King et al.

King et al. report parasite richness data for male guppies sampled from 18 populations in their native range in Trinidad. They found 21 species of parasites (most of them identified to the genus level) and show significant differences in parasite richness at the individual and population levels. Their data shows that individual size, snail richness and spatial distance among populations correlates with parasite richness. I think this paper will be of interest to the large community of researchers working on guppies. I found tables S9 and S3 particularly valuable, although I was a bit disappointed that many of the parasites were not identified to the species levels (I do recognize that doing so can be extremely challenging).

Major comments

-Selection

Throughout the manuscript, particularly in the conclusion, the authors make claims about the links between parasite richness and selection in the guppy system. Beyond the reasonable expectation that these different parasites (e.g. the most abundant or virulent ones) could influence selection of host resistance, other claims are highly speculative and should be clearly flagged as such. The data does not support these claims since no life-history, behavioural, physiological or other traits of the hosts were measured. Furthermore, the fact that parasite richness is higher than predator richness does not suggest that parasites are a stronger source of selection; I suggest removing such statements.

-Distance and connectivity

Connectivity and distance between populations are used interchangeably in the text (e.g. paragraph starting at L215). These two are not the same, particularly in streams. For example, given that the Northern Slope streams are all independent drainages they are not interconnected (no connectivity) and what the spatial distance between populations implies is not straightforward. The Caroni basin has instead all streams interconnected (although in a directional manner, which does not seem to be accounted for in the analyses) and distance between populations can have clear cut implications for parasite distributions. I suggest the authors clarify what they mean by distance and interconnectedness, and provide clear arguments for using these measures in streams that don't have a common freshwater tributary (Northern Slope). Furthermore, the authors should clarify how they measured distance, am I right to assume that they simply measured distance over a map using a straight line between populations? This would be problematic since it can be a poor reflection of direction of flow and the actual distance fish or parasites would have to travel among populations, and assumes topography plays no role in the movement of parasites and hosts.

-Sampling effort, snail and predator richness

The authors present correlations between parasite richness and predator and snail richness which form an important component of their rationale for the paper and their discussion (e.g. L65-68, L233-243). Nonetheless how the information on predator and snail richness was gathered is omitted (predators) or vaguely stated (snails). Was the sampling effort at all sites comparable? How exhaustive were the methods to detect these other species? Understanding this would help ease concerns when evaluating results such as those in Figure S2, where the regression seems to be strongly influenced by two sites with no snails, or Figure S3, comparing population-pair community dissimilarities in predators or parasites.

-Digenean instead of parasite correlations with snail richness

The authors formulate a hypothesis linking snail richness to overall parasite richness. It would be more adequate to present correlations between snail-transmitted parasites and snail richness only (the rationale here is clear, unlike mechanisms linking Trichodina, Camallanus or Gyrodactylus to snail richness).

-Parasite species identification

I suggest the authors explain further how different species of Digenea sp. 1-9 were identified (and possibly add images as supplementary materials). Knowing that there are 9 species present is of limited use, whereas researchers working in Trinidad could benefit from being able to distinguish among those species. Similarly, please clarify whether all Gyrodactylus were identified by location on the host or via microscopy? These methods differ substantially in reliability.

-Correlations

Given the nature of the study (field data collection) the results can only support correlations between certain variables and parasite richness, therefore the host size, snail richness and

distance among populations cannot “explain” parasite richness. This should be revised throughout the manuscript.

Minor comments

-I suggest the authors use the term “richness’ instead of “diversity” / “biodiversity”, particularly in the title and abstract, as this is the only measure of diversity in this study.

-L25 -Be specific about “effects of predators on guppy biology”

-L26 -Clarify how this is going to aid studies of natural selection.

-L37 -Clarify model organism for what.

-L84-85 -More information is needed – describe the protocol, sampling effort and what did ‘supplementing’ consisted on.

-L90 -‘Wallis’ not “Wallace”.

-L94 -Clarify what kind of variation.

-L112 -Physical distance can interpreted in many ways, please be very clear on how distance was measured.

-L210 -Clarify what wider range implies: diets, habitats?

-L250-251 -This claim is not supported by the data, and is not a claim that can be reasonably made based on differences in richness.

-L251-253 -This is highly speculative and overstates the findings.

-L258 -“RH” is not one of the co-authors or is it a typo? Please clarify (1) what was their contribution, and (2) how the fish used in this study were chosen out of the 10,000 that were potentially sampled. If RH is not one of the co-authors please properly acknowledge their contribution.

Supplementary Materials

-Information under the subheading “Sampling” explaining the rationale for using males and guppy collection procedure is better suited for the main text rather than as a supplementary material. While reading the manuscript I wondered why the authors had only used male guppies and whether the sampling protocol increased parasite transmission.

Author's Response to Decision Letter for (RSOS-191112.R0)

See Appendix A.

RSOS-191112.R1 (Revision)

Review form: Reviewer 2

Is the manuscript scientifically sound in its present form?

Yes

Are the interpretations and conclusions justified by the results?

Yes

Is the language acceptable?

Yes

Do you have any ethical concerns with this paper?

No

Have you any concerns about statistical analyses in this paper?

No

Recommendation?

Accept with minor revision (please list in comments)

Comments to the Author(s)

The manuscript by Mohammed et al. has improved considerably relative to the previous version. I consider my earlier concerns to have been sufficiently addressed. I have only some minor comments.

-I still consider that the use of "biodiversity" instead of simply 'richness' can be misleading and confusing to readers. Although the authors did measure parasite numbers of each species in individual hosts (and populations), they do not use them to estimate, for example, measures that account for evenness in the parasite communities. I also suggest that when describing the snail and predator communities the authors use the term richness instead of biodiversity.

-L75 - Doesn't 'Stress Coat' stimulate the production of mucus on guppies? Wouldn't this differentially remove ectoparasites (e.g. *Gyrodactylus* sp.) from your samples?

-L78 - If parasites were counted within 24h, how likely is that *Gyrodactylus* populations continued to grow on the sampled fish?

-L112 - "(i)" is not a reason for using physical distance, just a description of the characteristics of the system.

-L161-164 - Please explain how you corrected for multiple testing. Given the number of pairwise tests one could expect at least a couple of significant results per table.

-Figure 2 - I suggest that you describe the figure and add letters/stars to highlight significant differences but report the analyses in the main text instead of the caption. Also, replace "taxa" by 'species' or otherwise explain why you used other taxonomic level than species (applies to SI too).

Clarifications:

-L60 - clarify distance between populations

-L65-66 - Consider rewording (the same argument is better written in the supplementary information)

-L96 - Do the authors mean instead 'population nested within drainage'? "drainage nested within population" pops up in some other sections of the manuscript.

-L101 - ...presence/absence of parasite 'species'

-L104 - ...number of digenean parasite 'species'

-L154-155 - Meaning of "when including variation between populations" is unclear to me. Did you mean 'when we included population/locality as a factor ...'

-L186 - "...North Slope guppies are [in general] more..."

-L200 - Given the description in the methods (L93-94) it should be 'richness', not 'diversity'.

-L206 - Clarify "correlated" with what factor.

-L210 - "...to the number of parasites [of a given species] on fish"

Decision letter (RSOS-191112.R1)

05-Dec-2019

Dear Dr Lighten,

On behalf of the Editors, I am pleased to inform you that your Manuscript RSOS-191112.R1 entitled "Parasite diversity and ecology in a model species, the guppy (*Poecilia reticulata*) in Trinidad" has been accepted for publication in Royal Society Open Science subject to minor revision in accordance with the referee suggestions. Please find the referees' comments at the end of this email.

The reviewers and Subject Editor have recommended publication, but also suggest some minor revisions to your manuscript. Therefore, I invite you to respond to the comments and revise your manuscript.

- Ethics statement

- Data accessibility

If you wish to submit your supporting data or code to Dryad (<http://datadryad.org/>), or modify your current submission to dryad, please use the following link:
<http://datadryad.org/submit?journalID=RSOS&manu=RSOS-191112.R1>

- Competing interests

- Authors' contributions

AB carried out the molecular lab work, participated in data analysis, carried out sequence alignments, participated in the design of the study and drafted the manuscript; CD carried out the statistical analyses; EF collected field data; GH conceived of the study, designed the study,

coordinated the study and helped draft the manuscript. All authors gave final approval for publication.

- Acknowledgements

- Funding statement

Because the schedule for publication is very tight, it is a condition of publication that you submit the revised version of your manuscript before 14-Dec-2019. Please note that the revision deadline will expire at 00.00am on this date. If you do not think you will be able to meet this date please let me know immediately.

on behalf of Dr Michael Tobler (Associate Editor) and Kevin Padian (Subject Editor)
openscience@royalsociety.org

Associate Editor Comments to Author (Dr Michael Tobler):

We have received additional feedback from a previous reviewer. Based on their and my own reading, this manuscript is acceptable for publication with minor revisions. In addition to the feedback by the reviewer, please consider the following things:

- Figure 2: Split the panels into three different figures.
- Figure 2B: Flip the axes for increased readability.
- Figure 2C, y-axis label should read "Jaccard's". Also provide a unit for the x-axis.

Reviewer comments to Author:

Reviewer: 2
Comments to the Author(s)

The manuscript by Mohammed et al. has improved considerably relative to the previous version. I consider my earlier concerns to have been sufficiently addressed. I have only some minor comments.

-I still consider that the use of "biodiversity" instead of simply 'richness' can be misleading and confusing to readers. Although the authors did measure parasite numbers of each species in individual hosts (and populations), they do not use them to estimate, for example, measures that account for evenness in the parasite communities. I also suggest that when describing the snail and predator communities the authors use the term richness instead of biodiversity.

-L75 - Doesn't 'Stress Coat' stimulate the production of mucus on guppies? Wouldn't this differentially remove ectoparasites (e.g. Gyrodactylus sp.) from your samples?

-L78 - If parasites were counted within 24h, how likely is that Gyrodactylus populations continued to grow on the sampled fish?

-L112 - "(i)" is not a reason for using physical distance, just a description of the characteristics of the system.

-L161-164 - Please explain how you corrected for multiple testing. Given the number of pairwise tests one could expect at least a couple of significant results per table.

-Figure 2 - I suggest that you describe the figure and add letters/stars to highlight significant differences but report the analyses in the main text instead of the caption. Also, replace "taxa" by 'species' or otherwise explain why you used other taxonomic level than species (applies to SI too).

Clarifications:

- L60 – clarify distance between populations
- L65-66 – Consider rewording (the same argument is better written in the supplementary information)
- L96 – Do the authors mean instead ‘population nested within drainage’? “drainage nested within population” pops up in some other sections of the manuscript.
- L101 - ...presence/absence of parasite ‘species’
- L104 - ...number of digenean parasite ‘species’
- L154-155 – Meaning of “when including variation between populations” is unclear to me. Did you mean ‘when we included population/locality as a factor ...’
- L186 – “...North Slope guppies are [in general] more...”
- L200 – Given the description in the methods (L93-94) it should be ‘richness’, not ‘diversity’.
- L206 – Clarify “correlated” with what factor.
- L210 – “...to the number of parasites [of a given species] on fish”]

Author's Response to Decision Letter for (RSOS-191112.R1)

See Appendix B.

Decision letter (RSOS-191112.R2)

13-Dec-2019

Dear Dr Lighten,

It is a pleasure to accept your manuscript entitled "Parasite diversity and ecology in a model species, the guppy (*Poecilia reticulata*) in Trinidad" in its current form for publication in Royal Society Open Science. The comments of the reviewer(s) who reviewed your manuscript are included at the foot of this letter.

on behalf of Dr Michael Tobler (Associate Editor) and Kevin Padian (Subject Editor)
openscience@royalsociety.org

Appendix A

Associate Editor's comments (Dr Michael Tobler):

We have received the feedback from two reviewers. Overall, both agree that the paper is meritorious. However, one reviewer also provided some excellent feedback on how to potentially improve the current version of the manuscript. Pending satisfactory revisions, this manuscript can be suitable for publication in RSOS. I would also like to point out that a condition of publication is that authors make their supporting data, code and materials available.

Comments to Author:

Reviewers' Comments to Author:

Reviewer: 1

Comments to the Author(s)

The authors present a field study on parasite load in guppies - a, as they point out, so far understudied factor in guppy selection. They provide very detailed data on individual infections and population infestation and show compelling evidence for what influences parasite load in different populations. It becomes clear that parasites play a major role in guppy ecology. The manuscript is well written and provides a large amount of interesting data (mainly in the supplements). The statistical analyses seems appropriate and the conclusions drawn from the results are convincing. I can therefore recommend publication. For Figure 1 I would appreciate a legend explaining the colour code of parasites in the pie charts.

> We thank the reviewer for their positive comments. We have amended Fig 1 as suggested.

Reviewer: 2

Comments to the Author(s)

Review King et al.

King et al. report parasite richness data for male guppies sampled from 18 populations in their native range in Trinidad. They found 21 species of parasites (most of them identified to the genus level) and show significant differences in parasite richness at the individual and population levels. Their data shows that individual size, snail richness and spatial distance among populations correlates with parasite richness. I think this paper will be of interest to the large community of researchers working on guppies. I found tables S9 and S3 particularly valuable, although I was a bit disappointed that many of the parasites were not identified to the species levels (I do recognize that doing so can be extremely challenging).

Major comments

-Selection

Throughout the manuscript, particularly in the conclusion, the authors make claims about the links between parasite richness and selection in the guppy system. Beyond the reasonable expectation that these different parasites (e.g. the most abundant or virulent ones) could influence selection of host resistance, other claims are highly speculative and should be clearly flagged as such. The data does not support these claims since no life-history,

behavioural, physiological or other traits of the hosts were measured. Furthermore, the fact that parasite richness is higher than predator richness does not suggest that parasites are a stronger source of selection; I suggest removing such statements.

>We agree with the reviewer, and have toned down our currently untested hypothesis that parasites exhibit a stronger source of selection, and have made sure to caveat the hypotheses that parasites *may* be an important source of selection, being untested at this time.

-Distance and connectivity

Connectivity and distance between populations are used interchangeably in the text (e.g. paragraph starting at L215). These two are not the same, particularly in streams. For example, given that the Northern Slope streams are all independent drainages they are not interconnected (no connectivity) and what the spatial distance between populations implies is not straightforward. The Caroni basin has instead all streams interconnected (although in a directional manner, which does not seem to be accounted for in the analyses) and distance between populations can have clear cut implications for parasite distributions. I suggest the authors clarify what they mean by distance and interconnectedness, and provide clear arguments for using these measures in streams that don't have a common freshwater tributary (Northern Slope). Furthermore, the authors should clarify how they measured distance, am I right to assume that they simply measured distance over a map using a straight line between populations? This would be problematic since it can be a poor reflection of direction of flow and the actual distance fish or parasites would have to travel among populations, and assumes topography plays no role in the movement of parasites and hosts.

>We apologise for this initial confusion and have clarified our methods and rationale in the text, while also replacing "connectivity analysis" for "distance between populations".

-Sampling effort, snail and predator richness

The authors present correlations between parasite richness and predator and snail richness which form an important component of their rationale for the paper and their discussion (e.g. L65-68, L233-243). Nonetheless how the information on predator and snail richness was gathered is omitted (predators) or vaguely stated (snails). Was the sampling effort at all sites comparable? How exhaustive were the methods to detect these other species? Understanding this would help ease concerns when evaluating results such as those in Figure S2, where the regression seems to be strongly influenced by two sites with no snails, or Figure S3, comparing population-pair community dissimilarities in predators or parasites.

>Apologise for the lack of clarity. We have now described our sampling efforts for both predatory fish and snails. These were of equal effort at each site.

-Digenean instead of parasite correlations with snail richness

The authors formulate a hypothesis linking snail richness to overall parasite richness. It would be more adequate to present correlations between snail-transmitted parasites and snail richness only (the rationale here is clear, unlike mechanisms linking Trichodina, Camallanus or Gyrodactylus to snail richness).

>We agree and now present the analysis using digenean species (*Ascocotyle* sp., *Posthodiplostomum* sp., *Pygidiopsis* sp., *Saccocoelioides* sp., *Digenea* sp. 1-9).

-Parasite species identification

I suggest the authors explain further how different species of *Digenea* sp. 1-9 were identified (and possibly add images as supplementary materials). Knowing that there are 9 species present is of limited use, whereas researchers working in Trinidad could benefit from being able to distinguish among those species.

> Apologies for this initial oversight. Detailed species description were unfortunately beyond the scope of this study. SDK's experience led to the assumption that these represented different taxa based on the site that they infected in the fish, any particular distinctive attribute of size or eye spots, and/or if they were unique to a population. This information is now given in Table S2. We acknowledge that further work is needed to provide robust species delineations, as species ID in the field using light microscopy is difficult. In addition, we note in our Methods that lumping these taxa in fewer group did not alter the main outcomes of the study. We have provided the most detail currently available to us to aid future studies of digeneans in wild guppy populations.

Similarly, please clarify whether all *Gyrodactylus* were identified by location on the host or via microscopy? These methods differ substantially in reliability.

>Added. All were identified under microscope by SDK.

-Correlations

Given the nature of the study (field data collection) the results can only support correlations between certain variables and parasite richness, therefore the host size, snail richness and distance among populations cannot "explain" parasite richness. This should be revised throughout the manuscript.

>Agreed and corrected as suggested.

Minor comments

-I suggest the authors use the term "richness" instead of "diversity" / "biodiversity", particularly in the title and abstract, as this is the only measure of diversity in this study.

>Indeed we do measure parasite richness, but also measure the number of each parasite on each host, and compare metrics among populations on a spatial scale. As such we believe that biodiversity is appropriately used in the context when considering patterns across spatial scales.

-L25 –Be specific about "effects of predators on guppy biology"

>Added

-L26 –Clarify how this is going to aid studies of natural selection.

>Added

-L37 –Clarify model organism for what.

>Added.

-L84-85 –More information is needed – describe the protocol, sampling effort and what did 'supplementing' consisted on.

>Now clarified

-L90 –'Wallis' not "Wallace".

>Added

-L94 –Clarify what kind of variation.

>In the following sentence we clarify that this variation is 'population' and 'standard length'

-L112 –Physical distance can interpreted in many ways, please be very clear on how distance was measured.

>Added

-L210 –Clarify what wider range implies: diets, habitats?

>Apologies, this was a typo and is redundant in regards to the preceding "broader diet".
Removed.

-L250-251 –This claim is not supported by the data, and is not a claim that can be reasonably made based on differences in richness.

>Agreed and reworded.

-L251-253 –This is highly speculative and overstates the findings.

>Agreed and reworded.

-L258 –"RH" is not one of the co-authors or is it a typo? Please clarify (1) what was their contribution, and (2) how the fish used in this study were chosen out of the 10,000 that were potentially sampled. If RH is not one of the co-authors please properly acknowledge their contribution.

> Apologies. This was meant to read RSM – now first author

Supplementary Materials

-Information under the subheading "Sampling" explaining the rationale for using males and guppy collection procedure is better suited for the main text rather than as a supplementary

material. While reading the manuscript I wondered why the authors had only used male guppies and whether the sampling protocol increased parasite transmission.

>Agreed and corrected

Appendix B

Editor Comments

Figure 2: Split the panels into three different figures.

-Figure 2a-c now figures 2-4

- Figure 2B: Flip the axes for increased readability.

-Done.

- Figure 2C, y-axis label should read "Jaccard's". Also provide a unit for the x-axis.

-Done.

Reviewer comments to Author:

Reviewer: 2

Comments to the Author(s)

The manuscript by Mohammed et al. has improved considerably relative to the previous version. I consider my earlier concerns to have been sufficiently addressed. I have only some minor comments.

-I still consider that the use of "biodiversity" instead of simply 'richness' can be misleading and confusing to readers. Although the authors did measure parasite numbers of each species in individual hosts (and populations), they do not use them to estimate, for example, measures that account for evenness in the parasite communities. I also suggest that when describing the snail and predator communities the authors use the term richness instead of biodiversity.

-Agreed. 'Biodiversity' changed to 'species richness'

-L75 – Doesn't 'Stress Coat' stimulate the production of mucus on guppies? Wouldn't this differentially remove ectoparasites (e.g. *Gyrodactylus* sp.) from your samples?

-We did test this during collections by counting the number of dislodged *Gyrodactylus* in the small container that guppies were held in for n=20 samples across four populations. We found that the number of *Gyrodactylus* dislodges was negligible, and that stress coat/transportation to the lab had no effect on ectoparasites. We have now added this text to the methods.

-L78 - If parasites were counted within 24h, how likely is that *Gyrodactylus* populations continued to grow on the sampled fish?

Gyrodactylus parasites are viviparous and can reproduce one young every ~36 - 48h period, and individual parasites have a longevity of 10-14 days under optimal conditions. The error caused by the time delay counting these parasites is small and inevitable.

-L112 – “(i)” is not a reason for using physical distance, just a description of the characteristics of the system.

Agreed. Removed.

-L161-164 – Please explain how you corrected for multiple testing. Given the number of pairwise tests one could expect at least a couple of significant results per table.

- When examining the 7 species that were more abundant in the Caroni, we used a Bonferroni corrected alpha in the binomial tests (i.e. all p-values were $p < 0.00504$).

-Figure 2 – I suggest that you describe the figure and add letters/stars to highlight significant differences but report the analyses in the main text instead of the caption.

Also, replace “taxa” by ‘species’ or otherwise explain why you used other taxonomic level than species (applies to SI too).

- We have replaced taxa for species. We have not added letters or stars to the Figure 2 though and prefer to describe the findings in the caption instead.

Clarifications:

-L60 – clarify distance between populations

-Added.

-L65-66 – Consider rewording (the same argument is better written in the supplementary information)

- Agreed. We now removed comment on sampling from the SI to provide full detail in the main text.

-L96 – Do the authors mean instead ‘population nested within drainage’? “drainage nested within population” pops up in some other sections of the manuscript.

-Yes. Corrected.

-L101 - ...presence/absence of parasite ‘species’

-Added.

-L104 - ...number of digenean parasite ‘species’

-Added.

-L154-155 – Meaning of “when including variation between populations” is unclear to me. Did you mean ‘when we included population/locality as a factor ...’

-Yes. Corrected

-L186 – “...North Slope guppies are [in general] more...”

-Added.

-L200 – Given the description in the methods (L93-94) it should be ‘richness’, not ‘diversity’.

-Agreed.

-L206 – Clarify “correlated” with what factor.

-Typo. Removed.

-L210 – “...to the number of parasites [of a given species] on fish”]

-Added.